# Numerical Study on the Influence of Combined Rectification Facilities on the Flow in the Forebay of Pumping Station

Xiaobo Zheng [1], Pengli Zhang [2], Wenjing Zhang [1], Yue Yu [1] and Yaping Zhao [1,*]

1    Institute of Water Resources and Hydropower, Xi'an University of Technology, Xi'an 710048, China;
     zxb@xaut.edu.cn (X.Z.); 18309299001@163.com (W.Z.); 18871105987@163.com (Y.Y.)
2    Hanjiang to Weihe River Valley Water Diversion Project Construction Co., Ltd., Xi'an 710024, China;
     xazpl@163.com
*    Correspondence: zyp0168@163.com

**Abstract:** The flow pattern of the forebay of the pumping station has a considerable effect on the operating efficiency and stability of the pump unit. A good forebay flow pattern can enable the pump unit to improve efficiency and operating conditions. This study takes a large pumping station as the research object and considers two rectification schemes, namely, a single bottom sill and a "bottom sill + diversion pier". Without rectification facilities under different start-up schemes, the forebay flow pattern after the addition of rectification facilities is calculated, and the influence of single and combined rectification facilities is analyzed. Results show large-scale undesirable flow structures such as backflow and vortex in the forebay of the original design that without rectification facilities and uneven flow distribution occurs in the operating unit. The addition of a bottom sill in the forebay can control the central water beam from the water diversion pipe. The flow is divided to spread to both sides of the forebay and can be rectified twice after installing the diversion piers. The combined rectifier facility of "bottom sill + diversion pier" is beneficial to disperse incoming flow and make the flow distribution of each unit more uniform. The backflow and vortex inside the forepond are basically eliminated, and the flow state of the forepond is significantly improved.

**Keywords:** forebay; numerical simulation; rectification facilities; bottom sill; diversion pier





## 1. Introduction

The forebay of the pumping station is set up as a connection between the diversion channel and the water inlet pool to enable a smooth and even flow and thus provide good conditions for the pump unit. However, after the water flows into the forebay, adverse phenomena such as backflow and vortex may occur due to the influence of various factors, resulting in not only sedimentation in the forebay but also flow disorder in the water inlet channel of the pump unit. Thus, the operating efficiency of the pump decreases and severely affects the safe and stable operation of the unit [1–3].

For the study of the flow pattern of the forepool of the pumping station, in the past, the direct experiment mainly relied on the physical model, and the velocity measurement at the distribution point was mainly adopted by the current meter or particle image velocimetry (PIV) and laser Doppler anemometry (LDA), which show flow fields at character level [4,5]. With the development of computer technology, numerical simulation using computational fluid dynamics (CFD) has become a common research method, which can improve the efficiency of analysis and better study the flow field. Kim et al. [6] studied the flow distribution in the intake channel of the pump based on experiments and CFD technology to find out the cause of the eddy current in detail. At present, a large number of studies have been carried out on the flow state of the forebay. Harding et al. [7] established a mathematical model of the movement of the forebay in the velocity field and analyzed the internal flow field of the forebay by using an acoustic Doppler current profiler to quantify the errors caused by spatial changes in velocity. Amin et al. [8] believe that the efficiency of

the pumping station depends to a large extent on the structural design of the inlet pool, not only on the performance of the selected pump. Numerical and experimental studies were conducted on a rectangular inlet pool to predict the swirl angle and the formation of free-surface vortices, and the swirl angle and average tangential velocity estimated by CFD simulation were consistent with the experimental results. In the hydraulic model of seawater intake at the Aliveri power plant in Greece, Dimas and Vouros [9] studied the influence of cross flow in the front pool on the eddy current angle in the suction pipe of the pump and found that when the average cross flow velocity dropped below the critical value, the eddy current angle only depended on the shape of the front pool. Based on the above research, many scholars have carried out a great deal of analysis and research on the flow pattern. Zhang et al. [10] carried out experimental research on a lateral inflow pumping station at different water levels and found large-scale backflow areas in the forebay at the surface or the layers. Ying et al. [11] used a two-phase flow scheme to study the flow regime of a forward-influent forebay and found that the water flow separates at the side wall and large-scale backflows occur on both sides. For these flow characteristics of the forebay, domestic and foreign scholars have carried out research on improving the flow pattern of the forebay and carried out a lot of rectification measures for the forebay. Rtimi et al. [12] and Karami et al. [13] optimized the base splatter; Li et al. [14] and Luo et al. [15] added columns to the front pool and designed the columns; Ahmed et al. [16] and Zhou et al. [17] set different diversion pier layout methods; Xu et al. [18] used pressurized water plates for rectification of the front pool. Mi et al. [19] set a bottom sill in the forebay to eliminate the influence of eddy current and circulation generated by diffusion flow on pump performance and sediment deposition, and the research results can provide a theoretical basis for improving the flow pattern of the forebay and avoiding sediment deposition. Zhou et al. [20] analyzed the adverse flow state in the forebay of a certain lateral pumping station, such as large-scale backflow area and severe deflection of water inlet angle, and used various rectification measures to carry out numerical simulations; their results show that diversion piers can lead to a reasonable flow velocity area, the bottom sill can change the flow structure, and diversion walls can weaken the interference of backflow on the mainstream.

In addition, Nasr et al. [21] used various rectification measures to change the flow pattern of the front pool of the pumping station and found that when the parabola wall and part of the rectifier pier were well set, the flow pattern of the inlet pool was better and the uniformity of velocity distribution was improved. Yang et al. [22] studied and compared the rectifier flow pattern in the forepond, the uniformity of velocity distribution in the measured section, and the reduction rate of the eddy current area, and found that the combination scheme of the rectifier wall and the diversion wall had a good effect.

Existing literature has shown that adding bottom sills and diversion piers to the forebay can effectively improve the flow state, but it mainly focuses on the influence of a single rectification facility. Few studies examine the application of the combined rectification facility of the bottom sill and diversion piers. Therefore, the present study takes a large-scale pumping station as the research object and considers two rectification facilities: A single bottom sill and a "bottom sill + diversion pier". A three-dimensional (3D) numerical analysis is carried out on the flow with different numbers of units opened to examine the improvement effect of the bottom sill and the combination of the bottom sill and diversion pier on the flow. The findings can provide a reference for the design and renovation of rectification facilities for the same type of pumping stations.

## 2. Research Object

This study examines a large pumping station with five units, which are symmetrically arranged (4 working and 1 standby). The design flow rate of a single machine is 3.75 m$^3$/s. The forebay of the pumping station is 38.38 m long and is diverted by two steel pipes with a diameter of 2.6 m. The slope of the pool bottom is 0.049, the divergence angle of the forebay is 39.02°, and the inlet pool is 15 m long and 39.2 m wide. The center distance

between two adjacent pumps is 8.5 m, and the design water level is 19.5 m. The diameter of the trumpet tube is $D = 2.6$ m, the suspended height is 0.4 $D$, and the rear wall distance is 0.08 $D$. From top to bottom, these are units 1–5. The 3D model of the pumping station is shown in Figure 1.

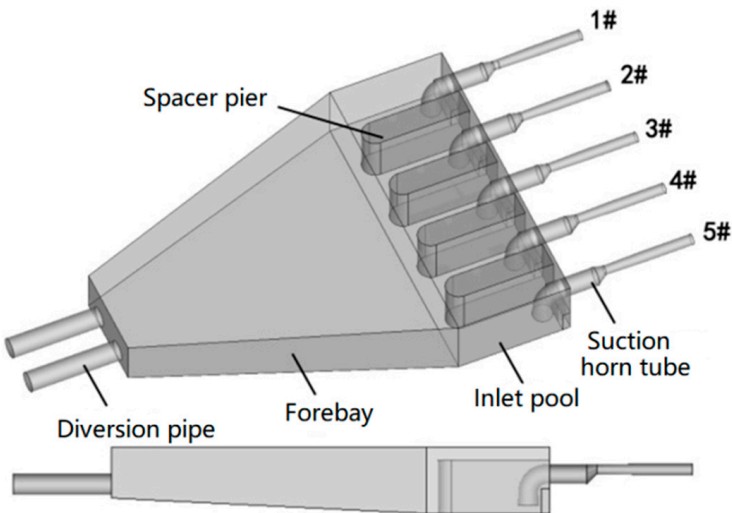

**Figure 1.** 3D model of pumping station.

## 3. Numerical Simulation

### 3.1. Turbulence Model

According to the hydraulic characteristics of the pumping station, the water movement in the forebay and the inlet pond of the pumping station is a turbulent flow with a large Reynolds number and a wide range of backflow and shedding. The RNG $k$-$\varepsilon$ turbulence model can manage flows with swirling, high strain rates, and large streamline bending, and thus this study adopts the RNG $k$-$\varepsilon$ turbulence model [23].

In this study, the CFX software is used to numerically simulate the flow in the forebay and inlet pond of the pumping station. The convection item of the turbulence model adopts a high-order precision format, and the convergence precision of each monitoring parameter is $10^{-4}$.

### 3.2. Calculation Region and Grid

Taking the water diversion pipe, forebay, water inlet pool, and water-absorbing trumpet for the calculation, the hexahedral structured grid is used to divide the different components into meshes, as shown in Figure 2.

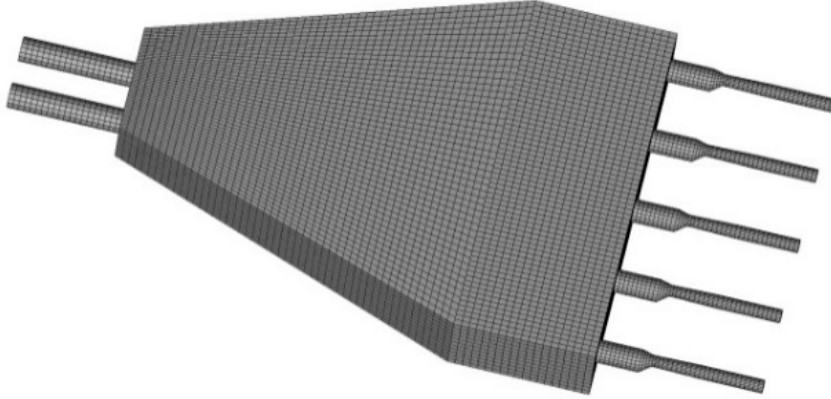

**Figure 2.** Calculation region and grid of pumping station.

Accurate numerical simulation results are obtained by verifying the calculation domain for grid independence. A total of six sets of grid schemes with different densities are generated, with the number of grids ranging from 1.76–3.05 million. The calculation results are shown in Figure 3.

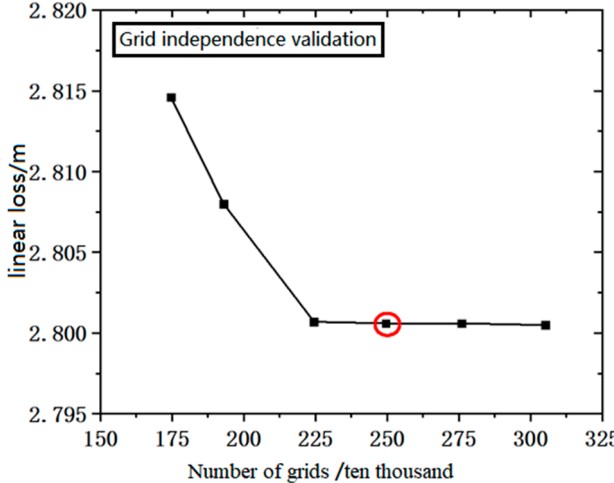

**Figure 3.** Grid independence validation.

The figure shows that when the number of grids increases from 1.76 to 2.24 million, the hydraulic loss in the calculation domain changes significantly, but when the increase is from 2.24 to 3.05 million, the hydraulic loss hardly changes, which meets the grid independence verification requirements. Therefore, 2.49 million meshes are selected for numerical simulation. The number of meshes for each flow-through component is shown in Table 1.

**Table 1.** Grid condition of each component.

| | Water Diversion Pipe + Forebay | Inlet Pool | Suction Trumpet | Total |
|---|---|---|---|---|
| Number of grids (10,000) | | 165.14 | 34.48 | 249.55 |

### 3.3. Boundary Condition

Inlet and outlet conditions: This study defines the inlet boundary as the inlet of the water diversion pipe, adopts the mass flow inlet boundary condition, and sets the design flow rate of a single pump as 3.75 $m^3$/s. The outlet is set at the outlet of the water-absorbing horn tube, and the static pressure outlet is set to 0 atm.

Wall conditions: All solid walls are smooth and non-slip (including side walls, bottom of the front pool, bottom of the water inlet pool, water diversion pipes, and water-absorbing trumpet pipes).

The surfaces of the forebay and the inlet pool are free liquid surfaces, treated symmetrically, assuming the use of the steel cover. The shear stress and heat exchange generated by air on the water surface are ignored.

### 4. Simulation Analysis of Forebay Flow in Original Scheme

The numerical calculation and analysis of the flow field of the research object is carried out by taking the operating conditions of Nos. 1, 2, 4, and 5 units in operation and No. 3 unit on standby as the calculation mode. Figure 4 shows the overall flow diagram of the pumping station, and after the water flow enters from the diversion pipe, the main flow is concentrated in the middle of the forebay. The flow velocity on both sides is relatively small. A large-scale backflow, distributed on both sides, and a vortex near the side wall are observed.

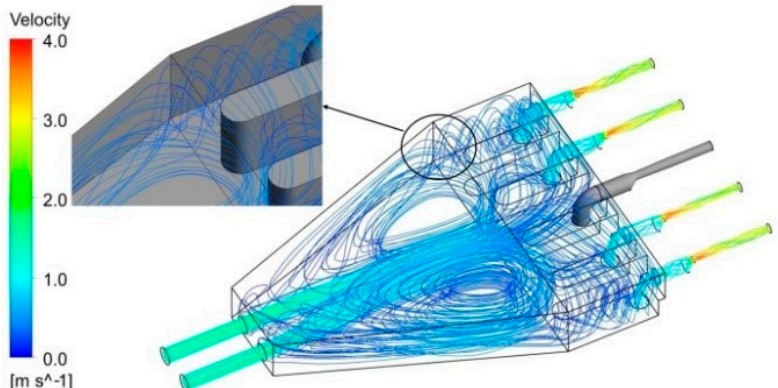

**Figure 4.** Flow pattern of the pumping station.

For further analysis of the internal flow of the inflow field of the pumping station, three sections were cut along the height direction of the forebay (the heights are: bottom layer $z$ = 13.5 m, middle layer $z$ = 16.5 m, surface layer $z$ = 19.5 m). The flow velocity vector diagrams of the three sections are shown in Figure 5. As the position moves down, the range of unfavorable flow patterns such as backflow and vortex expands and even goes deep into the pier of the inlet pool.

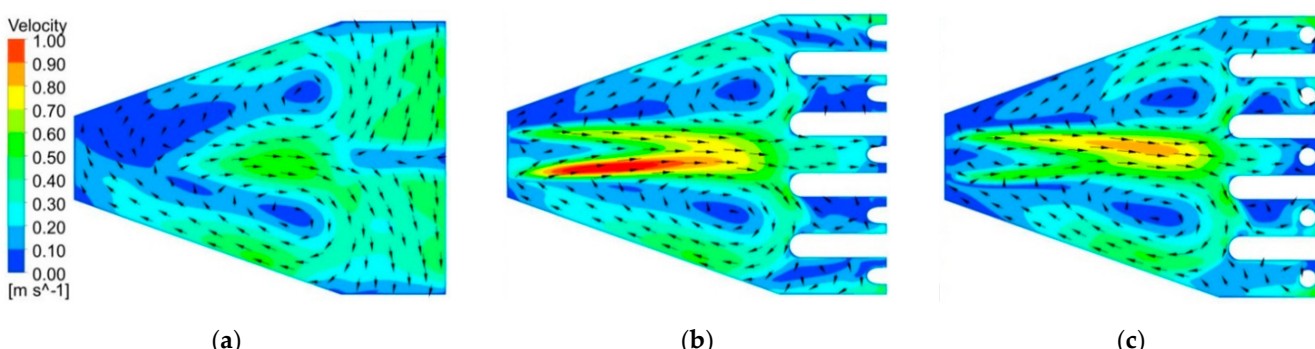

(**a**)　　　　　　　　　　　　　　　　　(**b**)　　　　　　　　　　　　　　　　　(**c**)

**Figure 5.** Streamlines of horizontal sections of forebay and pump sump: (**a**) Surface layer; (**b**) middle layer; (**c**) bottom layer.

Figure 6 shows the vorticity diagram of the bottom surface of the water inlet pool of the original scheme, and Figure 7 shows the vortex structure diagram under the water suction horn tube of each operating unit. Vortices of different scales form near and below the water-absorbing horns of each operating unit and at the head of the pier. The bottom vortex is clearly shown in Figure 7. The pier is formed by the impact of water flow, and the attached bottom vortices under the water-absorbing horns of units 1 and 5 are relatively strong.

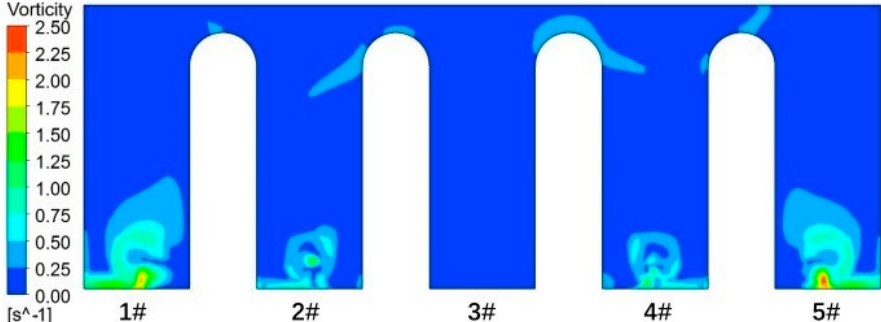

**Figure 6.** Vorticity distribution of pump sump.

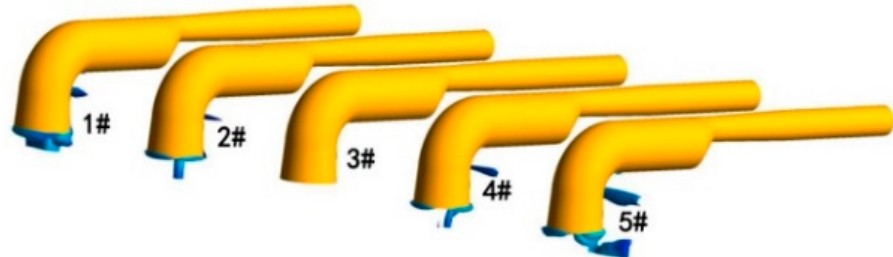

**Figure 7.** Vorticity of the original scheme operation unit.

From the above analysis, the reasons for the adverse flow conditions such as backflow and vortex in the forebay can be attributed to the following:

(1) The diffusion angle of the forebay is too large, close to the critical value of the pumping station design specification [24] (the forebay diffusion angle of the forward water pumping station is <40°).

(2) The forebay of the pumping station uses pressurized steel pipes to divert water, which causes high-speed water jets to form and impact the pier, thereby forming reverse water flow. This occurrence causes large-scale backflow, vortexing, and other undesirable flow patterns in the forebay.

## 5. Analysis of Influence of Rectification Facilities on Forebay Flow

*5.1. Rectification Facilities and Operating Conditions*

The principle of bottom sill rectification is to use appropriate engineering measures to cause the façade behind the sill to swirl, disrupt the plane backflow, and then use the diversion function of the diversion pier to homogenize the incoming flow. Thus, bad flow patterns such as backflow and vortexing are effectively eliminated. Based on the original design, this study proposes two rectification schemes, as shown in Table 2 and Figure 8.

**Table 2.** Rectification scheme.

| Program Number | Rectification Measures | Scheme Description |
|---|---|---|
| Scheme 1 | Bottom sill | Located at 10 m in front of the forebay, across the bottom, width × height: 1 × 1.5 m |
| Scheme 2 | Bottom sill + diversion pier | Data The diversion pier is located 7 m behind the bottom sill, and the distance between the heads of the two diversion piers is 8.7 m. The length × width × height: 10 × 1 × 5 m, and the included angle with the center line of the forebay is 18° |

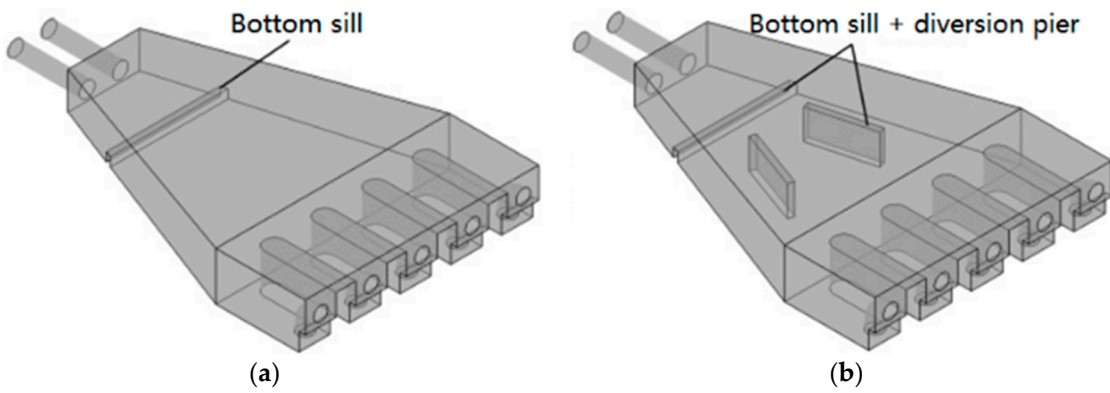

**Figure 8.** Structure diagram of rectification measures: (a) Scheme 1; (b) Scheme 2.

Given that the research object is the parallel operation of multiple units, this study determines the influence of different numbers of units on the flow state of the forebay of

the pumping station. The parallel operations of 3, 4, and 5 units are examined, and the operating conditions and start-up combinations are shown in Table 3.

**Table 3.** Operating conditions.

| Operating Conditions | Rectification Measures | Flow m³/s | Boot Group Number |
|---|---|---|---|
| 1 | | 11.25 | 1#, 3#, 5# |
| 2 | Original design scheme | 15 | 1#, 2#, 4#, 5# |
| 3 | | 18.75 | 1#, 2#, 3#, 4#, 5# |
| 4 | | 11.25 | 1#, 3#, 5# |
| 5 | Scheme 1 | 15 | 1#, 2#, 4#, 5# |
| 6 | | 18.75 | 1#, 2#, 3#, 4#, 5# |
| 7 | | 11.25 | 1#, 3#, 5# |
| 8 | Scheme 2 | 15 | 1#, 2#, 4#, 5# |
| 9 | | 18.75 | 1#, 2#, 3#, 4#, 5# |

*5.2. Analysis of the Influence of Rectification Facilities on Forebay Flow*

5.2.1. Comparative Analysis of Forebay Flow under Different Operating Conditions

Figure 9 is a vector diagram of the flow velocity in the forebay and inflow tank of the pumping station under various operating conditions. The velocity vector of each point in the flow field can be seen, so as to understand the velocity distribution of all parts in the flow field. Figure 10 is a vorticity diagram of the flow velocity in the forebay and inflow tank of the pumping station under various operating conditions, and the vorticity distribution in each region can be obtained.

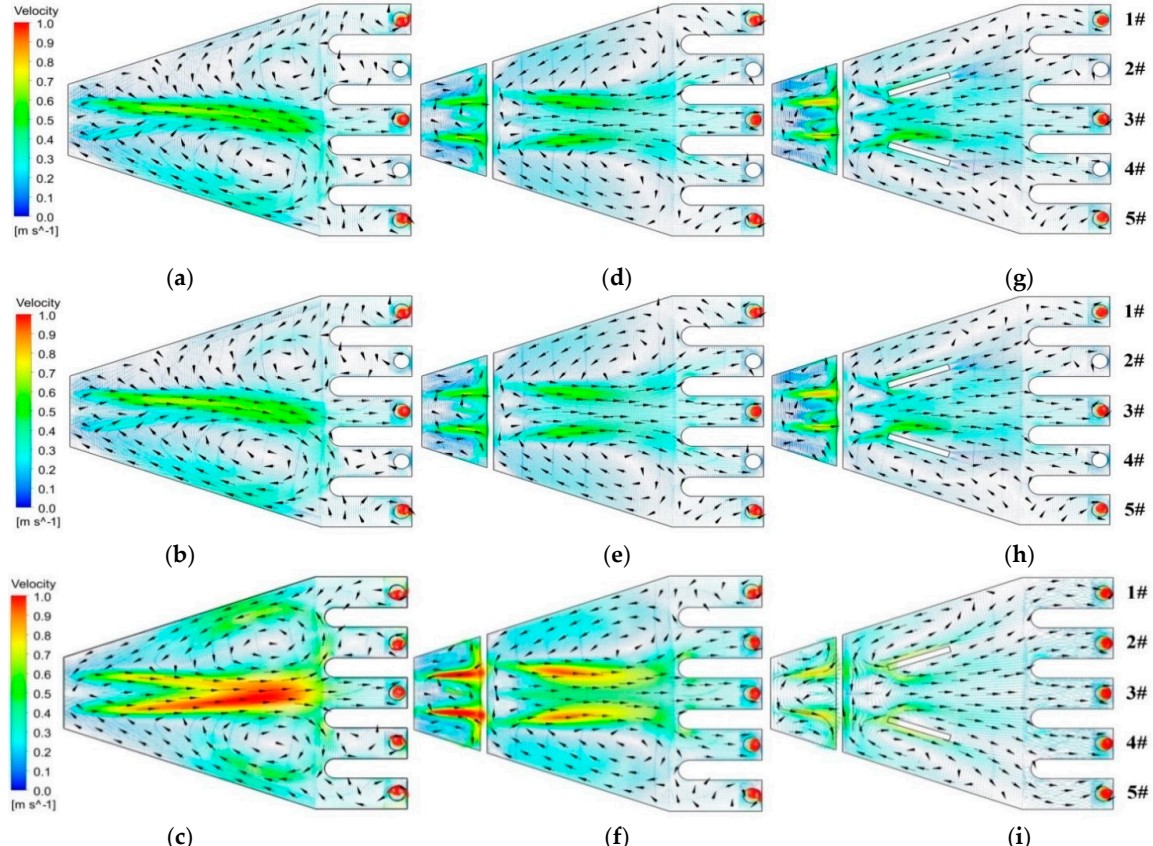

**Figure 9.** Velocity vector diagram of horizontal section (*z* = 13.5 m) of forebay and pump sump under different operating conditions: (**a**) Condition 1; (**b**) condition 2; (**c**) condition 3; (**d**) condition 4; (**e**) condition 5; (**f**) condition 6; (**g**) condition 7; (**h**) condition 8; (**i**) condition 9.

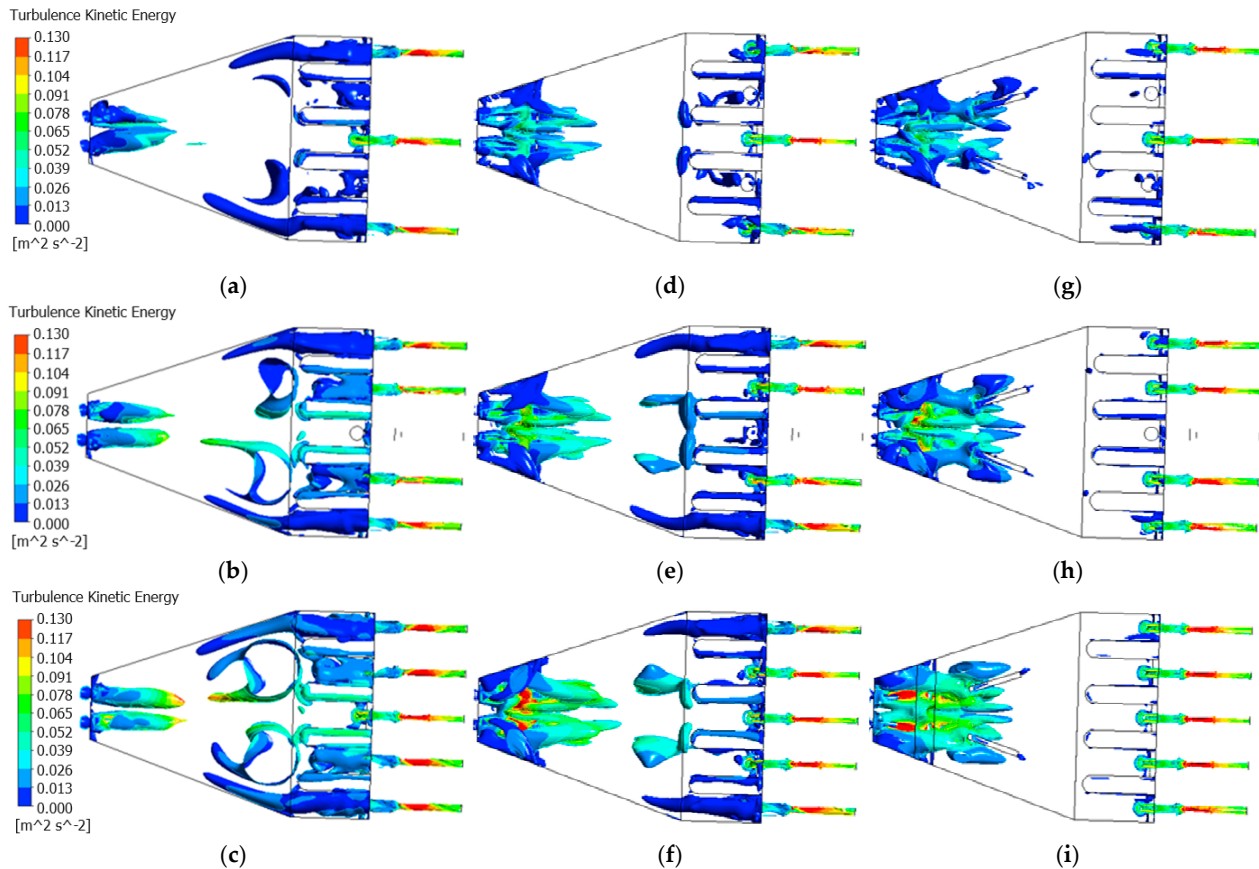

**Figure 10.** Vorticity diagram of horizontal section of front pool and inlet pool under different operating conditions: (**a**) Condition 1; (**b**) condition 2; (**c**) condition 3; (**d**) condition 4; (**e**) condition 5; (**f**) condition 6; (**g**) condition 7; (**h**) condition 8; (**i**) condition 9.

According to the analysis of the original design scheme of the forebay (i.e., working conditions 1–3 in Figure 9), when the pumping station starts 3, 4, and 5 units at the same time, a large-scale backflow occurs and is symmetrical on both sides of the forebay, and the scope of the spread goes deep to the pier of the inlet pool. Except for the No. 3 unit, the water inlet channels of the other operating units have a lateral side flow.

In view of the problems of the original scheme, the bottom sill is added to the forebay, and the water flow with large kinetic energy is intercepted by the bottom sill so that the water flow in the forepond is evenly distributed. Figure 9 Working Cases 4–6 show the vector diagram of flow velocity in the forebay section of Scheme 1. A bottom sill is added to the forebay, and the water flow enters through the water diversion pipe. The bottom sill diverts the central water beam from the water diversion pipe and diffuses it to both sides of the forward pool, which improves the diffusion of the water flow on the plane and disrupts the backflow. The improvement effect is mainly reflected in the water inlet channels of Nos. 2 and 4, which have improved horizontal side flows. The lateral side flow remains in the inlet channel of Nos. 1 and 3 units.

By comparing the original scheme with Scheme 1, it can be found that after adding the bottom sill to the original scheme, the flow line in the middle area of the forebay has tended to be stable, and the flow line in the inlet channel has also been improved, but there are still transverse flow measurements on both sides, so the diversion pier continues to be added to divert the incoming flow through the bottom sill and carry out secondary rectification. Figure 9 Working cases 7–9 show the cross-sectional flow velocity vector diagram of Scheme 2. Based on Scheme 1, diversion piers are added and carry out a secondary rectification of the water flow, further enhancing the diffusion of turbulent kinetic energy. The large-scale backflow areas on both sides of the forebay basically disappear,

and the flow of the water before entering the inlet pool is relatively stable. On the bottom section, the flow state of the water inlet pool is smooth, and the suction conditions of the water pump considerably improve.

Figure 10 is the comparison diagram of vorticity between the front pool and the inlet pool of the pumping station under different operating conditions. It can be seen from the vortex intensity distribution diagram of working conditions 1, 2, and 3 that when 3, 4, and 5 units are opened at the same time in the pumping station, large-scale vortices are generated at the inlet pool and the inlet channel of the unit. After adding the base sill rectification, it can be seen from the vortex intensity distribution diagram of working conditions 4, 5, and 6 that the scale of the vortices on both sides of the front pool is reduced, and the vortices at the inlet pool and the unit inlet channel are greatly improved. The vortex distribution diagram for working conditions 7, 8, and 9 is obtained after the diversion pier is added. In the diagram, the large-scale vortices at the inlet pond and the inlet channel of the unit basically disappear, and the flow pattern at the inlet of the unit is greatly improved. It can be seen from Figures 9 and 10 that, compared with the original scheme, the scale and quantity of adverse flow patterns such as backflow and vortex in the forebay in each operating condition of the two rectification schemes have been reduced, and the flow patterns have also been significantly improved. Therefore, the study shows that the combined rectification method with base sill and diversion pier has the best rectification effect in the scheme.

Figure 11 shows the vorticity diagram of the bottom surface of the inlet pool after the addition of rectification facilities when four units are turned on. By comparison with Figure 6, after the addition of rectification facilities, the range of vortex attached to the bottom of the operating unit decreases, but the single bottom sill increases the local (Nos. 2, 4 units) water flow vortices; and the addition of "bottom sill + diversion pier" in the forebay effectively improves the inflow conditions of the pump unit.

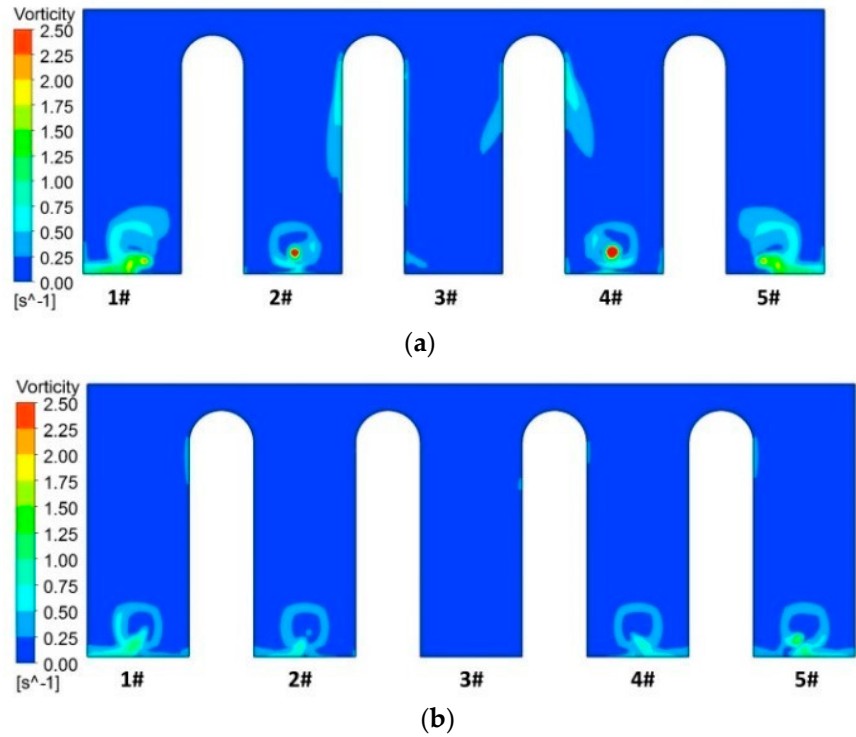

**Figure 11.** Vorticity distribution of pump sump: (**a**) Bottom sill; (**b**) bottom sill + diversion pier.

### 5.2.2. Comparative Analysis of Unit Flow Distribution

This study further analyzes the flow distribution among the operating units after the addition of rectification facilities. The flow distribution coefficient $\lambda$ is introduced, defined

as the ratio of the flow rate $Q_i$ of the suction horn tube corresponding to the operating unit to the design flow rate of a single pump $Q = 3.75$ m$^3$/s ($i$ represents the number of operating unit) and expressed as follows:

$$\lambda = \frac{Q_i}{Q} \tag{1}$$

The flow distribution coefficients of units under various operating conditions before and after the addition of rectification facilities are shown in Figure 12. As the $\lambda$ approaches 1, the flow distribution uniformity of the unit increases.

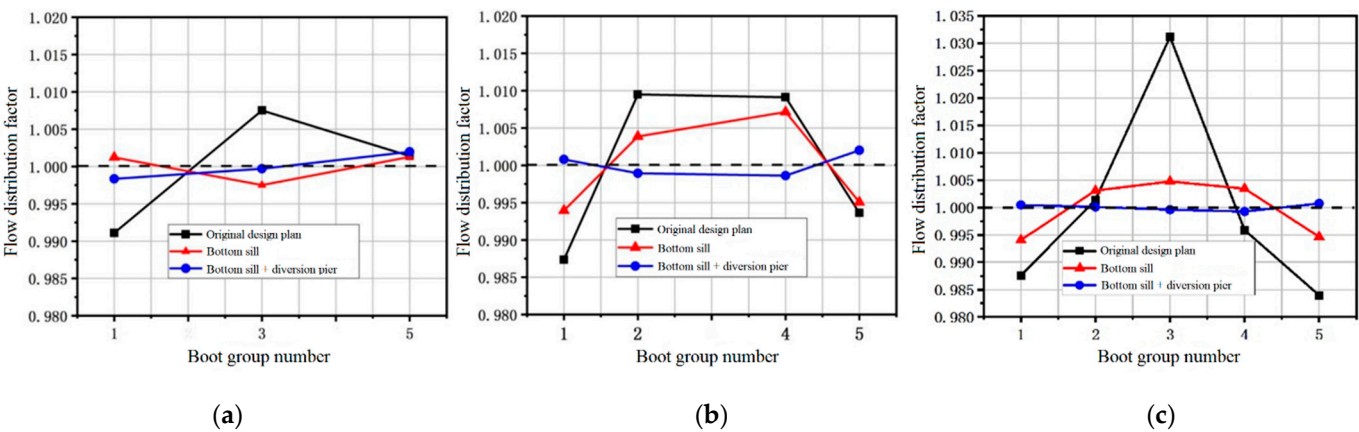

(**a**)　　　　　　　　　　　　(**b**)　　　　　　　　　　　　(**c**)

**Figure 12.** Flow distribution coefficient of operating units before and after rectification: (**a**) 3 units running in parallel; (**b**) 4 units running in parallel; (**c**) 5 units running in parallel.

The flow distribution coefficient curve shows that the original scheme has an uneven flow distribution, and the working flow of the unit operating in the middle is greater than that of units operating on both sides. In Scheme 1, a single bottom sill is set in the forebay, and the uneven flow distribution of each operating unit improves, but the effect is not apparent when four units are turned on at the same time. In Scheme 2, after the addition of the combined rectification facility of "bottom sill + diversion pier", the uneven flow distribution of each operating unit almost disappears.

### 5.2.3. Comparative Analysis of Flow Velocity Distribution Uniformity and Average Drift Angle

The flow velocity distribution uniformity and average drift angle at the inlet section of the water pump suction trumpet are important indicators to test the effect of the forebay renovation of the pumping station. This study introduces the velocity distribution uniformity $V_a$ and the cross-sectional average drift angle $\theta$ [25] and uses them as indicators to evaluate the actual effect of the rectification device.

The closer the cross-section flow velocity uniformity $V_a$ is to 100% and the average drift angle $\theta$ is closer to 0°, the more uniform is the axial flow velocity distribution of the water pump impeller inlet section. Thus, the inflow conditions of the water flow introduced into the pump unit improve, and the safe and stable pump operation and the operating efficiency of the pump device are also enhanced. The expression is as follows:

$$V_a = \left[ 1 - \frac{1}{\bar{u}_a} \sqrt{\frac{\sum \left( u_{ai} - \bar{u}_a \right)^2}{m}} \right] \times 100\% \tag{2}$$

$$\theta = \frac{\sum u_{ai} \left[ \arctan\left(\frac{u_{ti}}{u_{ai}}\right) \right]}{\sum u_{ai}} \tag{3}$$

In the above formula, $V_a$ is the uniformity of flow velocity distribution; $\theta$ is the average drift angle; $m$ is the number of units; $\overline{u}_a$ is the average axial velocity of the horn tube inlet; $u_{ai}$ is the axial velocity of each unit in section $i$; $u_{ti}$ is the lateral velocity of the $i$th calculation unit, and $u_{ti} = \sqrt{u_{wi}^2 + u_{ri}^2}$ ($u_{wi}$ and $u_{ri}$ are the tangential and radial velocities of the $i$th calculation unit, respectively).

Table 4 shows the flow velocity distribution uniformity and average deflection angle at the inlet section of the suction horn tube under various operating conditions before and after the addition of rectification facilities. Compared with Table 3 with the original scheme, after adding the rectification facilities, the uniformity of cross-sectional flow velocity distribution and the average drift angle have improved. This result shows that both optimization schemes can improve the unfavorable flow state in the forebay of the pumping station. However, the improvement of the second scheme is better achieved by adding a "bottom sill + diversion pier" combined rectification in the forebay, and the effect increases with the increase in the number of start-up units. With the addition of "bottom sill + diversion pier" combined rectification facilities, when Nos. 3, 4, and 5 units are turned on at the same time, the flow velocity distribution uniformity increases by 3.8%, 5.51%, and 7.46%, respectively.

**Table 4.** The uniformity and average drift angle of the inlet flow velocity distribution of the suction horn in various operating conditions.

|  |  | Original Scheme | Bottom Sill | "Bottom Sill + Diversion Pier" |
|---|---|---|---|---|
| Three units run in parallel | Velocity distribution uniformity $V_a$ | 73.58% | 76.20% | 77.38% |
|  | Mean drift angle $\theta$ | 18.21° | 16.89° | 15.42° |
| Four units run in parallel | Velocity distribution uniformity $V_a$ | 73.37% | 74.85% | 78.88% |
|  | Mean drift angle $\theta$ | 22.87° | 19.88° | 14.43° |
| Five units run in parallel | Velocity distribution uniformity $V_a$ | 70.52% | 75.62% | 77.98% |
|  | Mean drift angle $\theta$ | 21.09° | 20.33° | 15.30° |

## 6. Conclusions

Based on numerical simulation technology, this study analyzes the water flow state of the forebay of the original design scheme of the pumping station. According to the simulation results of the forebay, we propose a combined rectification scheme of two rectification schemes that influence the flow of the forebay of the pumping station under the condition of parallel operation of different sets of pumping stations.

1.　In the original scheme, the forebay of the research object has large-scale backflow, vortex, and other adverse flow structures when the units are turned on and running under the design working conditions. When multiple units run in parallel, uneven flow distribution occurs and the pumps have poor water inlet conditions, and this situation becomes more and more serious as the position moves down.

2.　The bottom sill is set in the forebay and causes the following benefits: Improves the centering of the mainstream, facilitates the diffusion of water flow on the plane, reduces the range of the forebay recirculation zone, and improves the uniformity of flow velocity distribution at the entrance of the suction horn pipe and the flow distribution uniformity of each operating unit. However, there are still backflow and vortices on both sides of the front pool, which need to be further improved.

3.　The combined facility of "bottom sill + diversion pier" is added to the forebay. The bottom sill diverts the central water beam from the water diversion pipe to spread to both sides of the forebay. The diversion pier then performs secondary rectification on the diverted forebay flow, which effectively improves the centering of the mainstream.

At present, the rectification effect of the combined rectifier facility on the forebay of the pump station is only the model stage, which proves that the technology is feasible,

and more experimental verification should be conducted in the future. At the same time, different factors, such as incoming flow speed and flow rate, will also have an impact. Finally, the size and location of the rectifier facility need to be specifically designed.

**Author Contributions:** Y.Z. and Y.Y. carried out the numerical simulations and analyzed the data; P.Z. and W.Z. conducted the experiment and wrote the first draft of the manuscript; W.Z. and X.Z. conceived and supervised the study and edited the manuscript; X.Z. and Y.Z. guided the experiment and the manuscript. All authors have read and agreed to the published version of the manuscript.

**Funding:** This work was supported by the Joint Foundation of Shaanxi [Grant No. 2019JLP-25].

**Data Availability Statement:** The data presented in this study are available on request from the corresponding author.

**Conflicts of Interest:** Author Pengli Zhang was employed by the company Hanjiang to Weihe River Valley Water Diversion Project Construction Co., Ltd. The authors declare that this study received funding from the company Hanjiang to Weihe River Valley Water Diversion Project Construction Co., Ltd. The funder had the following involvement with the study: research object, technical scheme, formal analysis, data curation. The other authors declare no conflicts of interest.

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
