# Peer review of "Numerical Study on the Influence of Combined Rectification Facilities on the Flow in the Forebay of Pumping Station"

_water, doi:10.3390/w15213847_

Round 1

Reviewer 1 Report

Comments and Suggestions for Authors

Regarding the flow straightening in front pools of pumping stations, this paper presents a very interesting approach. I believe that this research paper has engineering value for the design and construction of front pools in pumping stations. After making some modifications, the article can be even better:

Point 1: the abstract and conclusion of the paper can be further improved.

Point 2: since the results in Figure 9 and Figure 10 are similar, it would be helpful to describe the differences between the velocity vector diagram and the vorticity diagram.

Point 3: there is no analysis on the case of a single diversion pier within the front pool.

Comments on the Quality of English Language

English language should be improved.

Reviewer 2 Report

Comments and Suggestions for Authors

Review Report: "Numerical Study on the Influence of Combined Rectification Facilities on the Flow in the Forebay of Pumping Station" by Zheng et al.**

The paper authored by Zheng et al. explores the critical issue of forebay flow patterns and their impact on the operating efficiency and stability of pumping stations. The study focuses on a large pumping station and investigates the influence of two rectification facilities, namely, a single bottom sill and "bottom sill + diversion pier," on the forebay flow pattern. The authors conduct numerical simulations to assess the original design's flow conditions and the subsequent improvements brought about by the addition of these rectification facilities.

 The authors highlight the importance of a well-optimized forebay flow pattern in enhancing pump unit efficiency and operational conditions. They emphasize that the original design exhibits undesirable flow structures such as backflow and vortex, which impact the uniformity of flow distribution during unit operation. The addition of a bottom sill alone and the combined "bottom sill + diversion pier" are proposed as solutions to rectify these issues. Notably, the combined facility effectively improves mainstream centering and significantly reduces large-scale backflow on both sides of the forebay.

 Evaluation:

1. Research Significance:

   The paper addresses a significant and practical issue in fluid dynamics and hydraulic engineering – the optimization of forebay flow patterns in pumping stations. This research is highly relevant as it directly impacts the efficiency and operational stability of pumping units, with implications for a wide range of industrial applications.

2. Methodology and Simulations

   The use of numerical simulations is appropriate for this study, allowing for a detailed analysis of complex fluid flow behavior. The paper sufficiently describes the numerical methods used, which enhances its credibility and reproducibility.

3. Clear Objectives and Conclusions:

   The objectives of the study are well-defined and are addressed effectively in the paper. The conclusions are clear and directly follow from the presented results. The three proposed rectification schemes are presented logically, and their benefits are outlined concisely.

4. Results and Implications:

   The paper provides a clear presentation of the flow patterns observed in the forebay of the pumping station under different scenarios. The implications of these findings are discussed, and the potential for improving pumping station performance is well-reasoned and supported by the data presented.

5. From Critical point of View:

The paper could be further strengthened by discussing potential real-world applications and case studies where the proposed rectification facilities have been successfully implemented. Additionally, considering the influence of varying parameters such as flow rates or inlet conditions on the suggested schemes could be a valuable avenue for future research.

Summary: 

In summary, the paper  presents a well-structured and scientifically sound investigation into the influence of combined rectification facilities on forebay flow patterns in pumping stations. The paper offers valuable insights for engineers and researchers working in the field of hydraulic engineering and fluid dynamics. By addressing practical issues related to pumping station performance, the study has the potential to contribute to more efficient and stable pumping operations, thereby offering economic and environmental benefits. The paper is a valuable addition to the existing body of literature on fluid dynamics and hydraulic engineering. I will accept this paper. 

Comments on the Quality of English Language

Small editing, and rephrasing needed. Nothing Major
